# Glial Dysfunction in MeCP2 Deficiency Models: Implications for Rett Syndrome

**DOI:** 10.3390/ijms20153813

**Published:** 2019-08-05

**Authors:** Uri Kahanovitch, Kelsey C. Patterson, Raymundo Hernandez, Michelle L. Olsen

**Affiliations:** 1School of Neuroscience, Virginia Polytechnic and State University, Life Sciences I Building Room 212, 970 Washington St. SW, Blacksburg, VA 24061, USA; 2Department of Cell, Developmental, and Integrative Biology, University of Alabama at Birmingham, 1918 University Blvd., Birmingham, AL 35294, USA; 3Graduate Program in Translational Biology Medicine and Health, Virginia Tech, Roanoke, VL 24014, USA

**Keywords:** astrocytes, oligodendrocytes, microglia

## Abstract

Rett syndrome (RTT) is a rare, X-linked neurodevelopmental disorder typically affecting females, resulting in a range of symptoms including autistic features, intellectual impairment, motor deterioration, and autonomic abnormalities. RTT is primarily caused by the genetic mutation of the Mecp2 gene. Initially considered a neuronal disease, recent research shows that glial dysfunction contributes to the RTT disease phenotype. In the following manuscript, we review the evidence regarding glial dysfunction and its effects on disease etiology.

## 1. Introduction

Rett syndrome (RTT) is a rare, X-linked neurodevelopmental disorder that affects 1 in 10,000 females born worldwide. Despite the rarity of RTT, it is considered a leading cause of intellectual disability in females, accounting for an estimated 10% of all cases [1,2]. Patients with RTT demonstrate typical development from birth until 6 to 18 months of age, when motor, verbal, and cognitive skills begin to stagnate and regress, resulting in children exhibiting a range of symptoms typically including autistic features, intellectual impairment, motor deterioration, and autonomic abnormalities. The primary cause of RTT is genetic mutations of the *MECP2* gene located on the long arm of the X-chromosome at the Xq28 locus [3]. Although commonly believed to affect only females, RTT is most often the result of *de novo* mutations, which are prone to be of rapidly dividing sperm origin. Thus, although RTT has a probabilistic female predominance, affected males exist, and have likely been historically underrepresented due to the assumed lethality effect of an X-linked mutation in males [4].

The majority of RTT phenotypes stem from central nervous system (CNS) disruption. Rescue of Mecp2 in the central nervous system was generated by a conditional knock-in (cKI) using the nestin promoter. These animals displayed Mecp2-depleted peripheral tissues but normal protein levels in the developing CNS [5]. cKI mice did not display behavioral, locomotor, or early death trends seen in Mecp2 knockout (KO) models and had no demonstrable difference in respiratory behavior, cardiovascular recordings, or skeletal muscle morphology when compared to wild-type (WT) animals. These findings establish that CNS levels of MeCP2 are the greatest contributor to RTT symptoms typically associated with patient presentation. 

The CNS is comprised of a variety of cells that form the major tissues of the brain and spinal cord. In the case of RTT, the role of glial contribution to the symptoms has only recently been explored within the last 10 years. Initial studies in either post-mortem brains, primates, or mice found little to no expression of MeCP2/Mecp2 in glial cells [6,7,8,9,10]. Early evidence for glial involvement in RTT was established with the discovery that many glial transcripts involved in known neuropathology increased expression in the RTT brain [11]. With improved antibodies, MeCP2/Mecp2 was shown to be expressed in all glial cells in several different glial preparations [12,13,14,15,16,17]. Experiments restoring Mecp2 specifically to neurons were unable to prevent a Rett-like phenotype in mouse models [18], suggesting the possible involvement of other CNS cell types. In subsequent papers, it was shown that glia has a functional role in the development of RTT pathophysiology. The focus of this review will be to describe the contributions from the three major and distinct non-neuronal glial cell types. Namely, astrocytes, oligodendrocytes, and microglia will be discussed as they pertain to RTT pathophysiology.

## 2. Astrocytes in Rett Syndrome

Astrocytes are the most abundant glial cell type in the CNS, participating in roles that promote the survival and function of neurons by maintaining extracellular ionic concentrations, providing metabolic support, and responding to CNS injury. Astrocytes tile the brain neuropil, in which a single astrocyte is responsible for a non-overlapping domain, enveloping several neuronal cell bodies and contacting an estimated 2,000,000 synaptic sites in humans [19]. Astrocytes actively control dendritic growth, synaptogenesis, synapse number, synapse function, and synaptic plasticity [20]. With a large set of roles and responsibilities, structural or functional alterations of astrocytes can exacerbate pathological states. In 2009, it was first shown that *Mecp2^tm1.1Bird^* astrocytes exert non-cell-autonomous effects on WT neurons [15]. Research in the past decade has since revealed that astrocyte dysfunction in RTT is multi-faceted.

### 2.1. Differences in Gene Expression

In order to identify astrocyte-specific genes that are dysregulated in RTT, early culture studies utilized neonate animals to enable astrocyte isolation from other cell types. Astrocytes cultured from *Mecp2^tm1.1Bird/y^* mice express higher levels of astrocyte-specific genes, such as *Gfap* and *S100b* [21]. One study using primary cultured astrocytes from *Mecp2^tm1.1Bird/+^* and WT mice found 118 genes differentially regulated [22]. Another study using astrocytes cultured from *Mecp2^308/y^* mice found 257 affected genes relative to cultures from WT littermates, with little overlap with the previous study [23]. The genes identified were not of a particular gene cluster, but instead were involved in major cellular functions that are impaired in RTT, including cell–cell communication and cellular development. Thus, species and development-specific mechanisms, as well as sexual dimorphism and type of genetic model may all shape the response to Mecp2 deletion. It is important to note that until recently, astrocytes were cultured in the presence of fetal bovine serum (FBS). Serum is not typically present in the CNS and FBS demonstrates inter-vendor and inter-batch inconsistencies [24], which may contribute to variability between studies.

Astrocyte-specific gene changes can also be evaluated through various *Mecp2* KO and *Mecp2* knockdown (KD) models. Our lab has performed transcriptomic and proteomic analysis in the cortex of a global KO model, *Mecp2^tm1.1Jae/y^*. We found 391 differentially expressed genes, out of which 46 are astrocyte-specific, with pathway analysis suggesting deficits in genes associated with astrocyte maturation and morphology [25]. In a separate model, knocking down *Mecp2* from WT rats by siRNA injection was found to increase glial fibrillary acidic protein (Gfap) expression in the amygdala of female (as in [21]), but not male, rats. However, Mecp2 deletion did not affect the astrocyte-specific gene, *S100b*, nor *Gfap* expression in the hypothalamus of either sex [26], demonstrating the influence of sex and region on Mecp2 dysfunction. Sexual dimorphism is exhibited in another model of MeCP2 dysfunction – *MECP2* triplication syndrome. Astrocytes from male, but not female, transgenic mice of this model expressed elevated levels of Gfap [27].

### 2.2. Irregular Shape and Maturation

Astrocytes derive from neuronal stem cells (NSC) and increase in complexity during development to give rise to their mature form [28]. The morphological features of astrocytes are integral to the normal shape and functioning of the CNS [29]. Astrocyte maturation and morphology in RTT is likely dependent on the aforementioned differences in the RTT disease model, such as sexual dimorphism, regional heterogeneity, and slightly different culturing conditions. Astrocytes cultured from *Mecp2^tm1.1Bird/y^* mice do not show differences in growth rate [21], and induced pluripotent stem cells (iPSCs) derived from RTT patients differentiate to astrocytes at the same rate of iPSC controls [30], however, KD of *Mecp2* in mice primary astrocyte cultures via siRNA transfection causes lower astrocyte proliferation [12]. Astrocytes in *Mecp2^308/y^* mice show atrophic features, with fewer and poorly branched ramifications [31], and conditional KO of *Mecp2* at late juvenile stage causes hippocampal astrocytes to have significantly less complex ramified processes [32]. Alongside the body of literature demonstrating morphological aberration in RTT astrocytes, our lab has identified genes associated with astrocyte maturation and morphology that are differentially expressed in *Mecp2^tm1.1Jae/y^* global KO mice [25].

Astrocyte maturation parallels increased expression of microtubules, which coextend with intermediate filaments to form the main processes [29]. Astrocytes cultured from either male *Mecp2^308/y^* or male *Mecp2^tm1.1Bird/y^* display altered microtubule dynamics and growth rate, while cultured neurons from the same animals do not [33]. This astrocytic microtubule disruption causes impaired vesicular transport in astrocytes derived from human RTT iPSC [34]. In *Mecp2^308/y^* mice, astrocyte atrophy is reversed when mice are injected intracerebroventricularly (ICV) with bacterial cytotoxic necrotizing factor 1 (CNF1), an activator of RhoGTPases – a group of enzymes that induces dynamic changes in glial morphology [31]. Since astrocytic complexity correlates with the various functions executed by these cells [29], reductions in astrocyte morphologic complexity in RTT models suggest that there may be subsequent functional consequences.

### 2.3. Dysfunction in Metabolic Support

Astrocytes are multi-functional regulators of brain metabolism, and directly supply neurons with substrates for oxidative phosphorylation [35]. Therefore, astrocytic dysfunction may result in aberrant metabolic support in the brain. Brain magnetic resonance spectroscopy (MRS) of *Mecp2^308/y^* and *Mecp2^tm1.1Bird/y^* mice found significant metabolic anomalies, including a reduced amount of the sugar myo-inositol, which is an abundant osmolyte in glial cells and considered a putative glial marker [31,36]. Myo-inositol was normalized when mice were treated with CNF1 [31].

Before the discovery of the genetic background of RTT, early observations noted that RTT shares characteristics of mitochondrial encephalopathies and was first thought to be a mitochondrial disorder [37]. There are very clear indications for functional alterations of mitochondria in Rett syndrome leading to hypersensitivity to hypoxia and increased oxidative stress [38]. Astrocytes cultured from *Mecp2^tm.1.1Bird^* KO mice have more mitochondria, which display more oxidative behavior than WT astrocytes, though shape and mitochondrial membrane potential were not altered [39]. Mitochondria in rat astrocyte primary cultures that were treated with *Mecp2* siRNA had elevated expression of two proteins of the mitochondrial respiration chain from complexes I and III, along with lower activity of complexes II and III, without an increase in reactive oxygen species (ROS) or change in cell viability [40]. Using a genetically-engineered optical probe, roGFP1 (interestingly expressed mostly in glial cells), to detect oxidizing states, brain slices from *Mecp2^tm1.1Bird/y^* mice showed increased oxidation at pre-symptomatic stages [38]. Melatonin, which protects the electron transport chain by numerous means [41], represents an interesting future therapeutic direction for RTT.

One of the ways astrocytes support the energy needs of neurons is by releasing or ‘shuttling’ lactate to neurons, which is then used as a precursor for energy [42]. There is evidence that this function is preserved in RTT. In RTT human patients, brain MRS reveals no change in lactate [43]. Astrocytes in the ventral surface of the medulla oblongata or cerebral cortex in brains of either Mecp2^tm1.1Bird^ or *Mecp2^tm1.1Jtc^* mice show no differences in tonic or hypoxia-induced release of lactate compared to WT [44]. Thus, metabolic support appears to be somewhat preserved in RTT astrocytes, though mitochondrial abnormalities are apparent.

### 2.4. Dysfunction of Potassium and Neurotransmitter Homeostasis

Another classical role of astrocytes is maintaining ion and neurotransmitter homeostasis through the expression of various ion channels, exchangers, and transporters in their nanoscopic processes [45]. There is vast evidence demonstrating that these functions are abnormal in RTT astrocytes.

Astrocytes remove about 90% of all released glutamate in the CNS by transporters such as EAAT1 and EAAT2 [46]. After glutamate is transported into astrocytes, glutamate is converted by enzymes, such as glutamine synthetase (GS), into the precursor glutamine and recycled back to synapses for reconversion into active neurotransmitters [45]. Whether or not the glutamate clearance role of astrocytes is dysfunctional in RTT is not clear. Cerebrospinal fluid and brains of RTT patients contain high levels of glutamate [47,48]. However, another brain MRS study showed decreased levels of glutamine/glutamate in brains of *Mecp2^tm1.1Bird/y^* mice [36]. Additional studies found that astrocytes cultured from *Mecp2^tm1.1Bird/y^* brains had higher glutamate clearance due to a lack of negative feedback from glutamate transporter downregulation [21]. The explanation may stem from a sex-specific effect, as studies with *Mecp2* triplication syndrome models show that male primary astrocyte conditioned media (ACM) contained high concentrations of glutamate, while female derived ACM did not [27].

The data regarding potassium buffering in the Rett brain is lacking. Extracellular K^+^ is critical in defining the resting membrane potential of neurons, and its removal from the extracellular space is vital for maintaining homeostasis of the surrounding environment. Astrocytes are key mediators of extracellular K^+^ clearance [49]. An important protein for potassium homeostasis maintenance by both astrocytes [50] and oligodendrocytes [51] is the inwardly-rectifying potassium channel Kir4.1. The promotor of *Kcnj10*, the gene encoding Kir4.1, is developmentally regulated by changes in DNA methylation [52], and Mecp2 is known to directly bind this promotor [53]. One study reports an increase of Kir4.1 in the locus coeruleus of *Mecp2^tm1.1Bird^* mice [54]. On the contrary, our own findings demonstrate that Kir4.1 is downregulated across all brain regions in *Mecp2^tm1.1Jea/y^* mice, and in accordance with its role in potassium homeostasis, the mice exhibit high extracellular potassium [53].

### 2.5. Astrocyte Reactivity and Rett Syndrome

Astrocyte reactivity is a term coined for the morphological and functional response seen in astrocytes after CNS injury and other neurological diseases. This defensive reaction of astrocytes is conceivably aimed at handling acute stress, limiting tissue damage, and restoring homeostasis. There are many molecular and some morphological features of reactive astrocytes, the most classical of which are upregulation of the intermediate filament GFAP, inducing hypertrophy, and cellular proliferation, causing hyperplasia [55].

It is generally accepted that RTT is a neurodevelopmental disorder in which the brain fails to mature, as opposed to a neurodegenerative disease in which healthy brain cells deteriorate and die [56]. Therefore, it would be expected that RTT astrocytes would not be reactive, as there is halted development rather than an identifiable injury. Astrocytes in Rett patients were found to be occasionally mildly hyperplastic [57]. Spinal cord samples from Rett patients have a markedly increased amount of glial fibrils between nerve fibers [58], and gliosis was abundant in the cerebellum [59] and cortex [60]. A later review evaluating pathological changes in human Rett tissue indicate no consistent changes in GFAP expression, and when GFAP expression was increased, it was difficult to discern if this was a primary or secondary event [61]. Expression of genes related to astroglial reactivity, such as *Gfap Apoe*, *Hsp27*, and *Clu* were found to be increased in RTT brains [11]. GFAP/Gfap is also found to be upregulated in human solitary tract [62], primary cortical astrocytes from *Mecp2^tm1.1Bird^* mice [21], hypothalamus and amygdala of female, but not male, siRNA KD rat astrocytes [26], and astrocytes derived from RTT patient iPSCs [63] (for sexual dimorphism critique, see above and [26,27]). RNA sequencing and proteomics from the whole cortex in symptomatic *Mecp2^tm1.1Jae/y^* global KO mice revealed no evidence of reactive gliosis, but instead a downregulation of ‘typical’ markers of astrocyte reactivity, including *Gfap*, *Gap43*, *Vim*, *Hspb1*, and *Anxa3* [25]. In line with these studies, morphometric analysis revealed reduced Gfap positive area in the dentate gyrus and the corpus callosum of *Mecp2^308/y^* model mice relative to WT littermates [31]. In neurons and iPSCs, Mecp2 is implicated in *Gfap* regulation, Mecp2 binds to the promotor [63] and methylated exon 1 [64] of the *Gfap* gene and represses its expression.

Astrocyte reactivity occurs in response to brain inflammation. Reactive astrocytes express many receptors for pro-inflammatory factors, including cytokines IL-1β and TNF-α. In response to stimulation, astrocytic NF-κB is activated, and large amounts of pro-inflammatory cytokines and neurotoxins, such as NO, H_2_O_2_, NH_4_, and glutamate, are released, contributing to increased neuroinflammation and neuronal death [65]. This activation is enhanced in the LPS model of inflammation in mixed astrocyte-microglia primary cultures from *Mecp2^tm1.1Bird/y^* mice [66]. Despite the enhanced response, baseline release of neurotoxins is not altered [21,66]. This evidence suggests a compounding effect of primed astrocytes increasing their response to neuroinflammation, while at baseline RTT astrocytes appear less neurotoxic. Considerable research is needed to elucidate whether astrocytes in RTT are predisposed to respond to inflammation by activation, overexpress astrogliosis-related genes, or have a combinatory presentation that results in increased inflammation within certain contexts. However, it is clear that astrocytes do not appear to be classically reactive in RTT as is seen in many neuropathological contexts including injury, epilepsy and neurodegenerative disease.

### 2.6. Dysfunction of Astrocytic Support to Synaptogenesis and Dendritic Morphology

Astrocytes are active and crucial components of synaptogenesis, refinement, and maintenance. They secrete factors that affect pre-synaptic and post-synaptic elements, modulate synaptic plasticity, and play a role in synaptic elimination [67,68]. Therefore, astrocytic dysfunction in RTT may contribute to the abnormal neuronal morphology observed in RTT [61] via a non-cell autonomous mechanism. Indeed, primary WT neurons co-cultured with primary *Mecp2*-null astrocytes (both the *Mecp2^tm1.1Bird^* and the *Mecp2^tm1.1Jae^* mutations) are immature with stunted dendrites compared to neurons cultured with WT astrocytes [13,15]. Similarly, astrocytes derived from human RTT iPSCs and the ACM from these cells also have an adverse effect on WT mouse neurons [69]. Finally, ACM from *Mecp2*-siRNA-treated primary astrocytes induces reduced neuronal viability in WT dorsal root ganglion neurons [70]. These aberrant neurons also show significant functional changes. Neurons grown with iPSC-derived RTT astrocytes, for example, have lower miniature excitatory postsynaptic currents (mEPSCs) [69]. Importantly, restoring *Mecp2* specifically to astrocytes in *Mecp2^tm1.1Jae^* mice restores normal dendritic and synaptic morphology in vivo [71].

This aberrant neuronal dendrite morphology was hypothesized to stem from a lack of secretion of a morphogenic factor from mutant astrocytes or a neurotoxic agent produced by them, as ACM from mutant astrocytes did not support mature neuronal morphology [13]. However, there is data to refute these theories. ACM from highly pure RTT primary astrocyte cultures has no effect on neuronal morphology, pointing toward another cell type as a contaminant source of neurotoxicity from earlier studies [13]. This notion is supported by further studies implicating microglia in this process [72]. However, the question of how RTT astrocytes might exert apparent neurotoxic effects remains unanswered. Different methods of culturing astrocytes by use of magnetic cell sorting [53,73] or immunopanning [17] may be useful for these experiments.

Apart from supporting neurons, astrocytes also facilitate oligodendrocyte function by releasing growth factors that promote their differentiation from oligodendrocyte precursor cells (OPCs), promote maturation, and promote expression of genes that are involved in myelination [74]. ACM from *Mecp2*-siRNA-treated KD astrocyte cultures exerted a negative effect on OPC proliferation, and when mature oligodendrocytes were co-cultured with KD astrocytes, proteins associated with myelination, such as MBP and PLP, were downregulated [70]. Historically, the neuron-glia interaction has been a focus of interest in understanding the role of typical and dysfunctional astrocyte contribution to health. However, the growing evidence for glial-glial interactions elevates the role of astrocytes in healthy and disease states such as RTT.

### 2.7. Gliotransmission

Accumulating evidence supports the presence of a dynamic, bidirectional communication between neurons and astrocytes. Astrocytes express receptors to neurotransmitters and neuromodulators such as mGluR5, CB1, P2Y, GABA_B_, and many more. Activation of these receptors leads to an intracellular increase in (Ca^2+^)_i_, causing astrocytes to release various active substances, such as glutamate, ATP, D-serine, and other so-called gliotransmitters. This, in turn, activates neuronal receptors, affecting synaptic plasticity and modulation [75,76]. Controversy surrounds the concept of gliotransmission, specifically, groups have questioned whether gliotransmission occurs under physiological conditions or is simply an artifact of techniques used [77]. Extensive reviews regarding gliotransmission and this controversy can be read elsewhere [77,78]. Here, we will discuss observed differences in Ca^2+^ dynamics and secretory factor release in RTT models.

Using the double-patch clamp technique in the barrel cortex, it was observed that astrocyte depolarization or application of an agonist specific to PAR1 (protease-activated Gα_q_-coupled receptor, expressed more in astrocytes than neurons in the barrel cortex according to this study) induces increased synaptic currents in nearby neurons. In *Mecp2^tm1.1Bird/y^* mice, this effect is abolished. Heterozygous *Mecp2^tm1.1Bird/+^* females enable the study of variations in Mecp2 expression of astrocyte-neuron pairs, and through this mechanism, it was determined that expression of Mecp2 in astrocytes mediates this phenomenon, regardless of Mecp2 expression in neurons [79]. Mecp2-null astrocytes, either cultured, in brain slice, or in vivo, as well as astrocytes derived from RTT iPSC, display higher intrinsic intracellular Ca^2+^ activity, with Mecp2 expression rescuing aberrant Ca^2+^ activity. Specifically, Mecp2-null astrocytes are found to have higher ER Ca^2+^ load and higher baseline (Ca^2+^)_i_. This impaired Ca^2+^ dynamic stems from the elevated expression of TRPC4 and causes excessive activation of extrasynaptic NMDA receptors in neurons [80].

Gliotransmission has also been evaluated in detail through studies aiming to define the role of astrocytes in central respiratory regulation. There is amassing evidence that specialized astrocytes sense pH, P_CO2_, and P_O2_ in specific brainstem regions, and modulate neuronal networks via gliotransmission, ultimately influencing the homeostatic control of blood gases [81]. Astrocytes in the surface of the ventral medulla respond to changes in pH by increasing (Ca^2+^)_i_ and releasing ATP, which affects neuronal networks controlling the hypercapnic response [44,82,83,84,85]. Interestingly, this stimulus-induced elevation in [Ca^2+^]_i_ is impaired in *Mecp2^tm1.1Bird^* mice [44], and specific KO of Mecp2 in astrocytes using the cre-lox system also depresses the normal hypercapnic response [86].

Other types of molecules that astrocytes release include neurotrophic factors. One of the most well studied of these molecules is a brain-derived neurotrophic factor (BDNF/Bdnf). BDNF is regarded as the most characterized of the pro-survival trophic factors released by neurons, often viewed as critical in facilitating network connectivity due to its activity-dependent release at synaptic sites [87]. *Bdnf transcription* is regulated by Mecp2, suggesting a particular target of interest in neuronal maturation. Indeed, the methylation of the *Bdnf* gene increases with age [88]. BDNF/Bdnf is implicated in RTT, as patients and animal models have been recorded to have lower levels of *BDNF/Bdnf* transcripts [89,90]. Overexpression of *Bdnf* in Mecp2 deficient mice improves locomotor function, increases brain weight, and restores the electrophysiological activity of somatosensory pyramidal neurons [90].

Several studies have specifically addressed the expression and release of Bdnf from glial cultures. Bdnf is expressed in astrocytes, but its expression is dependent on developmental stage, location, and pathophysiology [91]. While cultured neurons express multiple variants of *Bdnf* transcripts, cultured astrocytes express mainly one transcript of *Bdnf*, *Bdnf VI*, which is more lowly expressed in neurons [92]. Bdnf released from astrocytes modulates synapses, but its role in synapse formation is debatable. It appears that reactive astrocytes or astrocytes that receive other external signals such as depolarization or metabotropic activation produce more Bdnf [91]. The results concerning the effect of Mecp2 deficiency on Bdnf signaling via astrocytes is inconclusive. Symptomatic *Mecp2^tm1.1Jae/y^* mice have less Bdnf in their brains [90]. On the other hand, Mecp2 deficiency in astrocytes cultured from both *Mecp2^tm1.1Bird/y^* and *Mecp2^tm1.1Jae/y^* causes an increase in Bdnf transcription [15], in line with astrocyte reactivity in RTT. A different study that used astrocytic primary cultures from rats found that *Mecp2* KD astrocytes express more, but secrete less, Bdnf [70]. Remarkably, neurons from *Mecp^308/y^* mice had more *Bdnf VI* transcript, but not in astrocytes [92]. Given records of patient deficiency in BDNF and studies demonstrating the regulatory role of Mecp2 on Bdnf production, it is imperative for future research to determine mechanisms that explain how the processing and release of BDNF are altered in RTT within different cellular populations.

### 2.8. Contribution of RTT Astrocytes to the Breathing Phenomenon

Specific restoration of Mecp2 in astrocytes of RTT models has demonstrated sufficiency to rescue neuronal morphology in vivo and ameliorate motor and anxiety-like behavior abnormalities [71]. One pathophysiological symptom, however, was completely reversed by restoring Mecp2 in astrocytes – the breathing abnormalities [71]. A variety of brainstem, midbrain, and even rostral brain areas are hypothesized to contribute to disordered breathing in RTT, and glia appears to play an important role in central respiratory control more broadly [81,93]. Hypercapnic conditions drive astrocytes in the medulla to respond with intracellular Ca^2+^ elevations to facilitate ATP release, increase respiratory activity, and aid in re-oxygenation [44]. In 10% CO_2_ hypercapnic conditions, astrocytes in *Mecp2^tm1.1Bird^* and *Mecp2^tm1.1Jtc^* mice fail to elicit any detectable Ca^2+^ response [44], though these results arguably lack specificity and may be the result of deficits in neuronal respiratory mechanisms. In vivo studies comparing the respiratory patterns of astrocytic versus neuronal *Mecp2* conditional KO mice found that tidal volume (VT), or volume of inhalation, is reduced in both groups when compared to controls [86]. With increasing CO_2_ concentrations, VT also rises for both groups, but does not reach control levels of respiration. Frequency of breath does not differ across groups, demonstrating that RTT models can maintain breathing rates during hypercapnic challenges, yet fail to appropriately respond to maintain adequate oxygenation. Though both neuronal and astrocytic *Mecp2* KO results in similar presentation, the depreciated VT response in the astrocytic model exceeds that of the neuronal KO [86], further strengthening the argument that astrocytic loss of Mecp2 has a role in RTT symptomology, with respiratory behavior primarily implicated. Interestingly, studies also suggest a particular role for astrocytes in not only the development but also the maintenance of hindbrain breathing pathways, as later postnatal loss of *Mecp2* in astrocytes is sufficient to cause disordered breathing in previously normal animals, but fails to disrupt existing hippocampal circuitry [71].

## 3. Microglia in Rett Syndrome

Microglia act as the resident immune cells in the brain, providing support through secretion of pro-survival molecules, refining synaptic connections, and surveying the parenchyma for threats [94]. There exists a heated debate as to the extent of microglial involvement in the pathophysiology of RTT, or even if RTT microglia are dysfunctional. Microglia from WT animals have detectable levels of Mecp2 protein and transcripts, though at lower levels than neurons and astrocytes [72]. Microglia-like cells derived from RTT patient iPSC are significantly smaller than wild-type cells [95]. Brains of symptomatic *Mecp2^tm1.1Bird/y^* mice have fewer microglia, reductions in microglia soma size, and less process branching [96,97]. A separate study showed that these changes are perhaps either region-specific or sex-specific. In heterozygous symptomatic *Mecp2^tm1.1Bird/+^* females, microglia numbers were not decreased in the hippocampus and striatum, but hippocampal microglia had fewer branching nodes, while striatal microglia branching resembled that of control animals [98].

Mecp2 regulates inflammatory gene transcription in response to TNF stimulation [99]. Therefore, it has been suggested that Mecp2 may be a critical multifunctional regulator of inflammatory immune responses in multiple cell types [99]. In support of this notion, the quantity of cytokines released in response to LPS application is increased in mixed astrocyte-microglia primary cultures from *Mecp2^tm1.1Bird^*-null mice [66,100]. Morphological analysis, RNA-sequencing, and microarray analysis of microglia in *Mecp2* global KO demonstrate patterns that suggest increased activation states and inflammatory cytokine release [96,101], which may also have a role in RTT pathophysiology. With disease progression in this model, microglia become activated and subsequently depleted [96].

It has been hypothesized that the ratio between two populations of microglia, the classically activated microglia (M1) and alternatively activated microglia (M2), is disrupted in RTT. The M1 population releases pro-inflammatory cytokines and the M2 population releases anti-inflammatory cytokines in addition to various growth factors, such as BDNF and IGF1 [102]. A disparity in the genes in each of these activation states was reported in female *Mecp2^tm1.1Bird/+^* mice. Genes associated with microglia activation were significantly enriched, supporting the theory that RTT microglia demonstrate functional changes associated with increased inflammation [101]. Treating *Mecp2* KO mice with an analog of IGF1, one of the M2-secreted growth factors, significantly extends their lifespan, improves locomotor function, ameliorates breathing patterns, and reduces irregularity in heart rate [103]. Human clinical trials with recombinant IGF1 (mecasermin) showed some improvement in cognitive and social test scores [104]. Another study investigating the transcriptome of microglia from female *Mecp2^tm1.1Bird/+^* mice found reduced expression of 10 heat shock proteins [101], suggesting that RTT microglia may respond to environmental cues differently than WT cells.

Interestingly, when in vitro hippocampal neurons are treated with microglia conditioned media (MCM) from *Mecp2^tm1.1Bird/y^* animals, neurons show decreases in immunofluorescence for dendritic markers (PSD95, Map2, Ac-TN) with additional reductions in both dendritic length and health [72]. Importantly, *Mecp2^tm1.1Bird/y^* astrocyte culture media transferred to neurons using the same paradigm does not result in excitatory synaptic deterioration, demonstrating an effect specific to microglial loss of Mecp2. Of particular relevance is the finding that MCM of KOs contains a nearly fivefold increase in glutamate levels compared to controls [72]. Pretreatment of neuronal cultures with AMPA and NMDA receptor antagonists, NBQX and MK801, prior to KO MCM exposure preserves synaptic marker immunostaining. Prolonged glutamate signaling and synaptic exposure consequently leads to excitotoxicity, often resulting in large-scale cellular death and the disruption of excitatory/inhibitory balance. The release of glutamate may further explain white and gray matter volume reductions and high rates of epilepsy observed in RTT patients. Elevated glutamate may result from dysfunction in microglial mitochondria. Mecp2 serves as a microglia-specific repressor of *Slc38a1* (Snat1) expression, a sodium-coupled glutamine transporter. Snat1 overexpression or Mecp2 KO reduces the number of microglia, causes mitochondrial dysfunction, and results in over-production of glutamate leading to excitotoxicity [105]. Feedback from RTT neurons to microglia may also exacerbate the neuroinflammatory response. *Mecp2^tm1.1Bird/y^* mice lacking the chemokine receptor CX_3_CL1, which mediates neuron-glia communication, are less phenotypic, have better survival, higher body weight, improved respiratory parameters, better motor function, larger neuronal and microglial cell soma, and higher microglia abundance than KO microglia with intact signaling. Their microglia also expressed higher levels of IGF1 [97]. These data collectively suggest that the epigenetic regulation of microglial function may affect RTT neurodevelopment via inflammatory pathways as well as mechanisms independent of inflammatory mediators [106].

One fascinating yet controversial study proposed immunotherapy as a therapeutic approach in animal models, by irradiating *Mecp2* KO mice to remove all host immune cells, followed by the introduction of fluorescently tagged bone marrow from WT mice in the hopes of establishing microglia with typical Mecp2 expression patterns. The intervention increased survival odds, weight gain, brain weight, and cell soma size when WT-to-KO was compared against KO-to-KO transplantation. Behavioral and neurological scores also improved across multiple tasks measuring locomotor activity, respiratory patterns, and anxiety [107]. Others within the field, however, have adamantly challenged findings from this study. Wiping out the immune system is bound to cause a host of changes, and it is not known how many microglia repopulate the brain after the procedure [108]. Research groups have been able to replicate the engraftment of bone-marrow-derived microglia into the brain parenchyma, but were not able to reproduce any of the improvements noted by the initial experiment [109].

Alternatively, to an active role for microglia in pathological RTT states, others suggest that the deleterious effects of microglia may stem from their typical role in synaptic refinement, as they mount larger responses against an increasing prevalence of immature neurons. Microglia have a demonstrated role in synaptic pruning and development by engulfing synapses recognized by neuronal-glial signaling [110]. Specific KO of *Mecp2* in microglia results in no differences in synaptic engulfment patterns at 110 days of age when compared to controls, while global *Mecp2* KO results in increased synaptic engulfment. *Mecp2* re-expression in microglia of *Mecp2^tm2Bird/y^* animals does not decrease synapse elimination [111], suggesting that microglia may simply be fulfilling their role as synaptic pruners, facilitating the development of meaningful neuronal circuits. However, the dysregulation of activity-dependent circuit formation may result in increased pruning events across postnatal development, which has significant implications for the evolving symptoms across the RTT lifespan. Future work will need to take into consideration the roles of microglial activation states and cell-cell interactions with the goal of developing novel therapeutic targets.

## 4. Oligodendrocytes and OPC in Rett Syndrome

Although oligodendrocytes and OPCs are the least studied of all glial cells with regard to their role in RTT, evidence of their pathophysiologic involvement does exist. Oligodendrocytes themselves express Mecp2 [17,112]. Brains of human RTT patients have decreased white matter [59] and noticeable axonopathy [113]. The abundant oligodendrocytic proteins Crystallin B and S100α13 are upregulated in RTT post-mortem brains [11]. RTT oligodendrocytes have abnormal membrane-bound lamellated inclusions [114]. In the *Mecp2^tm2Bird^* mouse model, levels of myelin-related proteins are abnormal [115]. Transcriptomic and proteomic analysis performed by our lab has identified 26 differentially expressed oligodendrocytic genes in *Mecp2^tm1.1Jae/y^* mice [25].

In order to target postmitotic neurons, Luikenhuis et al. re-expressed Mecp2 under the Tau promoter, which rescues brain and body weight deficits in affected RTT animals [116]. However, the Tau promotor was also found to be highly expressed in oligodendrocytes (and, to a lesser extent, in astrocytes) [17], suggesting that rescue of phenotypes seen by [116] stems partially from the restoration of Mecp2 to mature oligodendrocytes.

In order to elucidate the role of Mecp2 in oligodendrocytes, Nguyen et al. generated mice lacking Mecp2 specifically in oligodendrocyte lineage cells using cre recombinase under the prototypical OPC promoter *Cspg4* (NG2). The generated mice demonstrate slight hyperactivity and develop severe hindlimb clasping phenotypes similar to those observed across RTT models. Furthermore, restoration of *Mecp2* in OPCs of *Mecp2^tm2Bird^* mice increases lifespan and improves the locomotor deficits and hindlimb clasping in both sexes, males further benefitted from fully restored body weight [115]. Altogether, the emerging data regarding oligodendrocyte dysfunction in RTT suggest a specific role for the cell type in motor abnormalities and hand clasping phenotypes in RTT patients.

The exact details regarding the mechanism(s) by which Mecp2 deficiency leads to oligodendrocyte dysfunction are unknown. siRNA KO of *Mecp2* from rat oligodendrocyte primary cultures leads to increased expression of genes that promote oligodendrocyte differentiation and myelination. Specifically, it was determined that Mecp2 binds to the promoters of *Mbp* and *Plp* [117]. Mecp2, then, appears to play a role in oligodendrocyte maturation from OPCs. In contrast, *Mecp2*-mutated neuronal progenitor cells showed decreased oligodendrocyte differentiation [118]. Another siRNA co-culture model showed that *Nf155*, a gene associated with axo-glial interactions, is decreased in *Mecp2* KD oligodendrocytes, while the axo-glial genes *Caspr* and *Nrl1* are upregulated in Mecp2 KD neurons [70]. Therefore, it appears that Mecp2 uniquely regulates neuron-oligodendrocyte communication in a cell-specific manner.

## 5. Developmental Aspects of Glial Dysfunction in Rett Syndrome

Research evaluating the role of Mecp2 in the CNS has defined how genetic mutation can alter RTT developmental trajectory. The specificity of expression in the brain is a direct determinant of RTT characteristics. Clinical RTT symptoms follow a progression through somewhat distinct phases, a process that may be governed by changing expression patterns of MeCP2. Developmental studies in mice and humans have demonstrated increasing levels of MeCP2/Mecp2 with age, with neurons in initially developed structures - the spinal cord and brainstem - exhibiting the earliest expression [6].

Studies have demonstrated that Mecp2 is expressed in mouse embryonic glial cells and is critical for establishing cell fate. siRNA [12] and retroviral studies [119] in cultured mouse embryonic and neuronal progenitor cells, as well as experiments from zebrafish [120], have shown that Mecp2 promotes neuronal differentiation and suppresses gliogenesis. Additional studies have determined that the absence of Mecp2 has minimal effect on neurogenesis in embryonic cultured mouse cells but induces gliogenesis and imparts an immature electrophysiologic phenotype on neurons [121]. Mecp2 has long been proposed to suppress gliogenesis during development via action on a STAT-dependent pathway [122]. A likely mediator of this process, miR-124, increases in abundance with neurogenesis [123,124] and is thought to promote neural and suppress glial differentiation via inhibition of a STAT3 pathway [125]. Interestingly, Mecp2 appears to influence STAT3 signaling by promoting miRNA-124 production [126], suggesting that its expression may suppress region-specific gliogenesis during key developmental windows. Mecp2 also regulates expression of additional miRNAs linked to neurogenesis, and vice versa [127,128,129]. Whether or not these interactions also precipitate changes in glial cell differentiation has yet to be elucidated.

Mecp2 is known to directly bind a variety of glial-specific genes, such as *Gfap* [63], *S100b* [130], and *Kcnj10* [53], and demethylation of these genes varies based on a developmental timeline corresponding to gliogenesis [130]. Neuronal binding of Mecp2 to glial-specific promoters is also critical for maintaining cellular phenotype and integrity throughout development [64]. Indeed, studies of human iPSC lines generated from RTT patients demonstrate increased astrocyte differentiation from neural precursor cells due to reduced MeCP2 binding to *GFAP* [63]. A recent study indicated that *Mecp2*-null neural progenitor cells (NPC) from adult *Mecp2^tm1Bird^* female mice tend to be senescent and demonstrated a marked decrease in differentiation to astrocytes and oligodendrocytes compared to *Mecp2*-positive cells from the same animals. Here neuronal differentiation was higher in the *Mecp2-*mutated NPCs than their *Mecp2*-positive counterparts [118]. The senescence phenotype is recapitulated in mesenchymal stem cells from mice [131], in human patients [132], and in glial cell count observed in wild-type and heterozygous mice [98]. These confounding reports suggest that Mecp2-dependent mechanisms affecting glial differentiation and/or proliferation are complex and likely dependent upon the degree of protein deficit, brain region, species, and other environmental factors.

Notably, the above study [98] reported decreased microglial complexity in the hippocampus of female heterozygous RTT mice, indicating a regional effect on maturation/morphology likely to influence cellular function. Mecp2 deficiency affects many glial-specific genes that are typically developmentally regulated, including those coding for ion channels and transporters that play key homeostatic roles. For example, Mecp2 binds directly to the promoter of *Kcnj10*, which is typically developmentally demethylated corresponding to increased protein expression and potassium uptake, processes that are disrupted in *Mecp2* mutant animals [52,53]. Cell culture experiments indicate a critical role for Mecp2 in promoting oligodendrocyte survival and myelination [70], which has clear implications for axonal maturation and development of white matter tracts in the brain. Broadly, these changes in network connectivity affect all stages of development, indicating that the consequences of Mecp2 depletion may have longstanding impacts on both glial and neuronal maturation and morphology [13,15,72].

A variety of Mecp2 depletion and rescue experiments demonstrate that this protein is important for CNS development as well as maintenance. Embryonic models of Mecp2 depletion recapitulate key features of RTT. Mecp2 depletion later in adulthood also results in significant deficits and mortality [133], while reactivation of Mecp2, including in glial cells alone, after symptom development results in phenotypic improvement without complete resolution [71,134,135,136]. These combined studies support the notion that RTT is a neurodevelopmental disease as well as a disease of cellular and network maintenance, with glia playing a key role.

## 6. Conclusions

Though the genetic determinants of RTT are largely known, treatments remain limited, likely due to the complex regulatory nature of MeCP2/Mecp2 across many cell types and throughout postnatal CNS development. The focus of current treatment strategies, pharmaceutical and otherwise, is to manage serious health concerns and comorbidities such as nutritional deficits, epilepsy, and motor deterioration. Though the technology for gene therapy has not yet advanced to the point at which it can be used in RTT, several other therapeutic approaches are being developed, some of which directly or indirectly target glial function.

Clinical trials focusing on the reestablishment of BDNF signaling are promising leads for immediate future treatments. Animal studies in *Mecp2* KO animals with simultaneous Bdnf overexpression have marked improvements in locomotion and autonomic dysfunction [90]. Similarly, pre-clinical research evaluating the effects of BDNF mimetic peptides have demonstrated reversal of apneas and respiratory abnormalities commonly associated with RTT [137]. Current trials evaluating therapeutic effects of fingolimod and copaxone, both FDA approved immunosuppressives typically used to treat MS, may improve patient conditions by increasing BDNF levels and modulating microglial activity [138,139].

Another therapeutic avenue that restores astrocytic shape is the use of activators of RhoGTPases, such as cytotoxic necrotizing factor 1 (CNF1). Use of this molecule in *Mecp2^308^* neonates dramatically reverses the evident signs of atrophy in mutant astrocytes and induces improved brain metabolism, glial integrity, and bioenergetics [31]. Additionally, restoration of microtubule stability in RTT astrocytes may prove viable as a therapeutic intervention. Relatively low weekly doses of Epothilone D, a microtubule-stabilizing drug, partially reverses the impaired exploratory behavior in *Mecp2^308/y^* male mice [34].

An alternative approach is to use compounds that specifically target reactive *Mecp2* mutant glia. Nance et al. studied the effects of dendrimer-conjugated N-acetylcystein (DNAC), an antioxidant prodrug that simultaneously decreases pro-inflammatory cytokine release. The dendrimers cross the blood-brain barrier (BBB) and accumulate in activated microglia and astrocytes [140]. *Mecp2^tm1.1Bird^* receiving DNAC show longer lifespan and an improvement in disease scores [66].

Although the viability of data regarding microglia replacement using bone marrow transplant [107] is highly contested, therapeutics targeting microglia in RTT remain exciting. Regulation of the microglial activation state to promote anti-inflammatory response may also be achievable pharmacologically with widely available drugs like minocycline [141]. Utilization of microglia M2-secreted anti-inflammatory factors, such as IGF1, may also promote improved control of inflammatory processes in the CNS. Both IGF1 and GPE (a peptide containing the first 3 amino acids of IGF1) are able to partially rescue the neuronal deficits caused by mutant RTT astrocytes [69].

In designing effective treatment strategies for RTT patients, it is imperative to understand the appropriate mechanisms of drug delivery. Astrocytes are critical to the maintenance of BBB properties under normal and pathological conditions [142]. Functional changes in the BBB in RTT have not been thoroughly evaluated, thus limiting our knowledge regarding the potential efficacy of drugs that rely upon movement through the BBB for CNS delivery. Given the evidence of glial dysfunction in RTT patients and animal models, it is crucial to assess whether BBB dysfunction may be a pathological hallmark capable of impairing therapeutic efforts.

Continued scientific investigation of glial involvement in RTT is key to determining additional treatment targets of interest. Through continued efforts in scientific advancement to inform clinicians, drive innovative treatments, and inform families affected by Rett syndrome, we may yet improve the lives of hundreds of thousands of people across the globe.

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
