# Peer review of "Glial Dysfunction in MeCP2 Deficiency Models: Implications for Rett Syndrome"

_ijms, 2019, doi:10.3390/ijms20153813_

Round 1

Reviewer 1 Report

Kahanovitch et al. provide an overview on the role of the glia in the development of RTT pathophysiology. In detail, authors describe the contributions from the three major and distinct non-neuronal glial cell types (i.e astrocytes, oligodendrocytes, and microglia). 

Although the paper is well-written and the authors highlight the proposed issues in a comprehensive way a major issue have to be addressed. 

Authors omit to mention a recent paper on the matter. This study analyzed the neural stem cell (NSC) differentiation by identifying the percentage of stem cells, precursors, and mature cells in the samples from RTT and wild-type mice. Authors noticed an increase in the percentage of neurons (MAP2+) and a decrease in astrocytes (GFAP+) and oligodendrocytes (01+). Interestingly, in progenitors and differentiated cells expressing mutated Mecp2, authors observed a very significant increase in senescent cells compared with the control cells (see Alessio N et al, 2018, PMID: 29563495).  The interesting association between RTT and senescence has been identified by some research groups (see Squillaro T et al. PMID: 22357617 and 31117273; Ohashi M et al, 2018, PMID: 29742391).

Author Response

We thank the reviewer for his important comment. We completely missed the article that the Reviewer mentions. Three papers regarding the differentiation of neuronal stem cells (NSC) in RTT have showed a decreased neuronal differentiation and increased differentiation into glia cells (Namihira@Taga 2004, Setoguchi@Nakashima 2006 and Andoh-Noda@Okano 2015). Another paper (Smith@Blue 2018) showed decreased differentiation to glia. The articles that Reviewer #1 brought up support the latter finding, and also add the possibility that the decreased glial differentiation is because of NSCs becoming senescent. In light of this insight, we revised the 2nd paragraph in page 10 to include this papers. We have also added “…Mecp2-null neuronal progenitor cells showed decreased differentiation to oligodendrocytes…” in the oligodendrocyte section (page 9, lines 431-433).

Reviewer 2 Report

This is a manuscript which reviews the physiopathology and clinical manifestations of Rett syndrome. 

The review is very interesting and well structured and written. 

There is only one point of interest, in my opinion, which the authors did not take into consideration: the role that high doses of melatonin can have in the improvement of Rett patients. Melatonin is a strong antioxidant, anti-inflammatory, and plays very important roles on the mitochondrial function, even is able to partially substitute the function of factors I and III from the respiratory mitochondrial chain when they are mutated or absent (mainly in the case of factor I). In fact, we treated a young girl, aged 6, with Rett syndrome, administering to her IGF-I and melatonin (50 mg/day) during 6 months, besides specific neurorehabilitation, and at discharge most of the typical signs of Rett syndrome had disappeared; even, the girl began to eat by herself, to watch videos, and began to speak. Therefore, in my opinion, the role of melatonin in this syndrome, together with IGF-I administration, should be analyzed.    

Author Response

We thank the reviewer for his insight. The main problem is that while the protective effect of melatonin on the electron transport chain complexes has been studied, there is no evidence in literature if this protective effect happens in RTT (melatonin has been used in RTT to treat sleep disorders), and no research has been made whether melatonin help mitigate oxidative stress in RTT glia cells. As this is an interesting new line of investigation, we have added “Melatonin, which protects the electron transport chain by numerous means (Hardeland 2017), is an interesting future therapeutic direction for RTT.”

Round 2

Reviewer 1 Report

Authors properly addressed the issues moved.

Neverthless, authors have to correct "Mecp2-null neuronal progenitor cells" with "Mecp2-mutated neuronal progenitor cells" when they refer to reference 118.

Author Response

We thank the reviewer for his comment. We have thoroughly combed the manuscript to check for grammar mistakes and various spelling errors and inconsistencies